# HYPE: Human eYe Perceptual Evaluation of Generative Models

**Sharon Zhou**[*]**, Mitchell L. Gordon**[*]**, Ranjay Krishna,**
**Austin Narcomey, Durim Morina, Michael S. Bernstein**
Stanford University
{sharonz, mgord, ranjaykrishna, aon2, dmorina, msb}@cs.stanford.edu

## Abstract

Generative models often use human evaluations to determine and justify progress. Unfortunately, existing human evaluation methods are ad-hoc: there is currently no standardized, validated evaluation that: (1) measures perceptual fidelity, (2) is reliable, (3) separates models into clear rank order, and (4) ensures high-quality measurement without intractable cost. In response, we construct Human eYe Perceptual Evaluation (HYPE), a human metric that is (1) *grounded* in psychophysics research in perception, (2) *reliable* across different sets of randomly sampled outputs from a model, (3) results in *separable* model performances, and (4) *efficient* in cost and time. We introduce two methods. The first, $HYPE_{time}$, measures visual perception under adaptive time constraints to determine the minimum length of time (e.g., 250ms) that model output such as a generated face needs to be visible for people to distinguish it as real or fake. The second, $HYPE_\infty$, measures human error rate on fake and real images with no time constraints, maintaining stability and drastically reducing time and cost. We test HYPE across four state-of-the-art generative adversarial networks (GANs) on unconditional image generation using two datasets, the popular CelebA and the newer higher-resolution FFHQ, and two sampling techniques of model outputs. By simulating HYPE's evaluation multiple times, we demonstrate consistent ranking of different models, identifying StyleGAN with truncation trick sampling ($27.6\%$ $HYPE_\infty$ deception rate, with roughly one quarter of images being misclassified by humans) as superior to StyleGAN without truncation ($19.0\%$) on FFHQ.

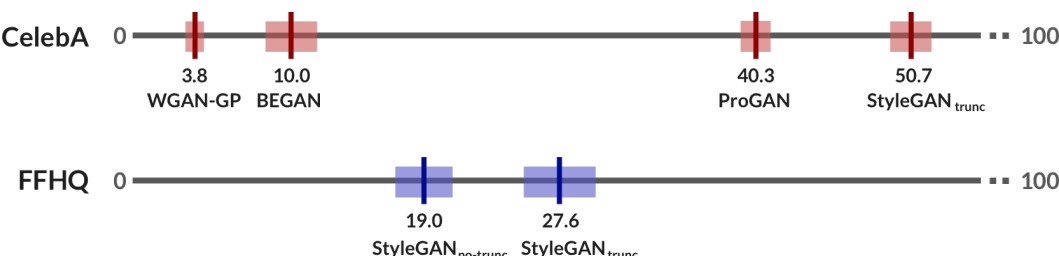

Figure 1: Our human evaluation metric, HYPE, consistently distinguishes models from each other: here, $HYPE_\infty$ scores compare StyleGAN, ProGAN, BEGAN, and WGAN-GP on CelebA, and StyleGAN with and without truncation trick sampling on FFHQ.

## 1 Introduction

Historically, likelihood-based estimation techniques served as the de-facto evaluation metric for generative models (Hinton, 2002; Bishop, 2006). But recently, with the application of generative models to complex tasks such as image and text generation (Goodfellow et al., 2014; Papineni et al., 2002), likelihood or density estimation grew no longer tractable (Theis et al., 2015). Moreover, for high-dimensional problems, even likelihood-based evaluation has been called into question (Theis

---

[*]Equal contribution.

et al., 2015). Consequently, most generative tasks today resort to analyzing model outputs (Rössler et al., 2019; Salimans et al., 2016; Denton et al., 2015; Karras et al., 2018; Brock et al., 2018; Radford et al., 2015). These output evaluation metrics consist of either automatic algorithms that do not reach the ideals of likelihood-based estimation, or ad-hoc human-derived methods that are unreliable and inconsistent (Rössler et al., 2019; Denton et al., 2015).

Consider the well-examined and popular computer vision task of realistic face generation (Goodfellow et al., 2014). Automatic algorithms used for this task include Inception Score (IS) (Salimans et al., 2016) and Fréchet Inception Distance (FID) (Heusel et al., 2017). Both have been discredited for evaluation on non-ImageNet datasets such as faces (Barratt & Sharma, 2018; Rosca et al., 2017; Borji, 2018; Ravuri et al., 2018). They are also much more sensitive to visual corruptions such as salt and pepper noise than to semantic distortions such as swirled images (Heusel et al., 2017). So, while automatic metrics are consistent and standardized, they cannot fully capture the semantic side of perceptual fidelity (Borji, 2018).

Realizing the constraints of the available automatic metrics, many generative modeling challenges resort to summative assessments that are completely human (Rössler et al., 2019; Salimans et al., 2016; Denton et al., 2015). These human measures are (1) ad-hoc, each executed in idiosyncrasy without proof of reliability or grounding to theory, and (2) high variance in their estimates (Salimans et al., 2016; Denton et al., 2015; Olsson et al., 2018). These characteristics combine to a lack of reliability, and downstream, (3) a lack of clear separability between models. Theoretically, given sufficiently large sample sizes of human evaluators and model outputs, the law of large numbers would smooth out the variance and reach eventual convergence; but this would occur at (4) a high cost and a long delay.

In this paper, we present HYPE (HUMAN EYE PERCEPTUAL EVALUATION) that addresses these criteria in turn. It: (1) measures the perceptual fidelity of generative model outputs via a **grounded** method inspired by psychophysics methods in perceptual psychology, (2) is a **reliable** and consistent estimator, (3) is statistically **separable** to enable a comparative ranking, and (4) ensures a cost and time **efficient** method through modern crowdsourcing techniques such as training and aggregation. We present two methods of evaluation. The first, called $HYPE_{time}$, is drawn directly from psychophysics literature (Klein, 2001) and displays images using adaptive time constraints to determine the time-limited perceptual threshold a person needs to distinguish real from fake (Cornsweet, 1962). The $HYPE_{time}$ score is understood as the minimum time, in milliseconds, that a person needs to see the model's output before they can distinguish it as real or fake. Small $HYPE_{time}$ scores indicate that model outputs can be identified even at a glance; large scores suggest that people need to dedicate substantial time and attention. The second method, called $HYPE_\infty$, is derived from the first to make it simpler, faster, and cheaper while maintaining reliability. It measures human deception from fake images with no time constraints. The $HYPE_\infty$ score is interpretable as the rate at which people mistake fake images and real images, given unlimited time to make their decisions.

We demonstrate HYPE's performance on unconditional generation of human faces using generative adversarial networks (GANs) (Goodfellow et al., 2014). We evaluate four state-of-the-art GANs: WGAN-GP (Gulrajani et al., 2017), BEGAN (Berthelot et al., 2017), ProGAN (Karras et al., 2017), and the most recent StyleGAN (Karras et al., 2018). First, we track progress across the years on the popular CelebA dataset (Liu et al., 2015). We derive a ranking based on perception ($HYPE_{time}$, in milliseconds) and error rate ($HYPE_\infty$, as a percentage) as follows: StyleGAN (439.4ms, 50.7%), ProGAN (363.7ms, 40.3%), BEGAN (111.1ms, 10.0%), WGAN-GP (100.0ms, 3.8%). A score of 500ms on $HYPE_{time}$ indicates that outputs from the model become indistinguishable from real, when shown for 500ms or less, but any more would start to reveal notable differences. A score of 50% on $HYPE_\infty$ represents indistinguishable results from real, conditioned on the real training set, while a score above 50% through 100% represents hyper-realism in which generated images appear more real than real ones when drawn from a mixed pool of both. Next, we test StyleGAN trained on the newer FFHQ dataset (Karras et al., 2018), comparing between outputs generated when sampled with and without the truncation trick, a technique used to prune low-fidelity generated images (Brock et al., 2018; Karras et al., 2018). We find that outputs generated with the truncation trick (363.2ms, 27.6%) significantly outperforms those without it (240.7ms, 19.0%), which runs counter to scores reported by FID.

HYPE indicates that GANs have clear, measurable perceptual differences between them. HYPE produces identical rankings between $HYPE_{time}$ and $HYPE_\infty$. We also find that even the best eval-

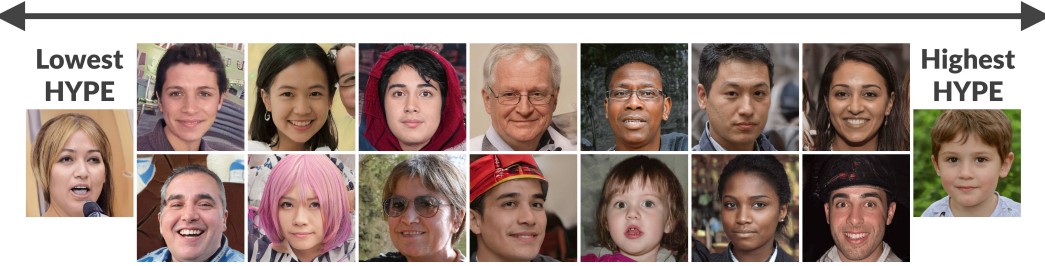

Figure 2: Example images sampled with the truncation trick from StyleGAN trained on FFHQ. Images on the right exhibit the highest HYPE scores, the highest human perceptual fidelity.

uated model, StyleGAN trained on FFHQ and sampled with the truncation trick, only performs at 27.6% HYPE$_\infty$, suggesting substantial opportunity for improvement. Finally, we show that we can reliably reproduce these results with 95% confidence intervals using 30 human evaluators at $60 in a task that takes 10 minutes. While important measures, we do not focus on diversity, overfitting, entanglement, training stability, and computational and sample efficiency of the model (Borji, 2018; Lucic et al., 2018) and instead aim to construct the gold standard for human perceptual fidelity.

We deploy HYPE as a rapid solution for researchers to measure their generative models, requiring just a single click to produce reliable scores and measure progress. We deploy HYPE at https://hype.stanford.edu, where researchers can upload a model and retrieve a HYPE score in 10 minutes for $60. Future work would extend HYPE to adapt to other generative tasks such as text generation or abstractive summarization.

## 2    HYPE: HUMAN EYE PERCEPTUAL EVALUATION

Model creators can choose to perform two different evaluations and receive two different scores: the HYPE$_{time}$ score, which gathers time-limited perceptual thresholds to measure the psychometric function and report the minimum time people need to make accurate classifications, and the HYPE$_\infty$ score, a simplified approach which assesses people's error rate under no time constraint. HYPE displays a series of images one by one to crowdsourced evaluators on Amazon Mechanical Turk and asks the evaluators to assess whether each image is real or fake. Half of the images are drawn from the model's training set (e.g., FFHQ or CelebA), which constitute the real images. The other half are drawn from the model's output. We use modern crowdsourcing training and quality control techniques to ensure high quality labels (Mitra et al., 2015).

### 2.1    PERCEPTUAL FIDELITY GROUNDED IN PSYCHOPHYSICS: HYPE$_{TIME}$

Our first method, HYPE$_{time}$, measures time-limited perceptual thresholds. It is rooted in psychophysics literature, a field devoted to the study of how humans perceive stimuli, to evaluate human time thresholds upon perceiving an image. Our evaluation protocol follows the procedure known as the *adaptive staircase method* (Cornsweet, 1962) (see Figure 3). An image is flashed for a limited length of time, after which the evaluator is asked to judge whether it is real or fake. If the evaluator consistently answers correctly, the staircase descends and flashes the next image with less time. If the evaluator is incorrect, the staircase ascends and provides more time.

This process requires sufficient iterations to converge on the minimum time needed for each evaluator to sustain correct guesses in a sample-efficient manner (Cornsweet, 1962), producing what is known as the *psychometric function* (Wichmann & Hill, 2001), the relationship of timed stimulus exposure to accuracy. For example, for an easily distinguishable set of generated images, a human evaluator would immediately drop to the lowest millisecond exposure. However, for a harder set, it takes longer to converge and the person would remain at a longer exposure level in order to complete the task accurately. The modal time value is the evaluator's perceptual threshold: the shortest exposure time at which they can maintain effective performance (Cornsweet, 1962; Greene & Oliva, 2009).

HYPE$_{time}$ displays three blocks of staircases for each evaluator. An image evaluation begins with a 3-2-1 countdown clock, each number displaying for 500 ms. The sampled image is then displayed for the current exposure time. Immediately after each image, four perceptual mask images are rapidly displayed for 30ms each. These noise masks are distorted to prevent visual afterimages and further sensory processing on the image afterwards (Greene & Oliva, 2009). We generate masks from the test images, using an existing texture-synthesis algorithm (Portilla & Simoncelli, 2000). Upon each submission, HYPE$_{time}$ reveals to the evaluator whether they were correct.

Image exposure times fall in the range [100ms, 1000ms], which we derive from the perception literature (Fraisse, 1984). All blocks begin at 500ms and last for 150 images (50% generated, 50% real), values empirically tuned from prior work (Cornsweet, 1962; Dakin & Omigie, 2009). Exposure times are raised at 10ms increments and reduced at 30ms decrements, following the 3-up/1-down adaptive staircase approach. This 3-up/1-down approach theoretically leads to a 75% accuracy threshold that approximates the human perceptual threshold (Levitt, 1971; Greene & Oliva, 2009; Cornsweet, 1962).

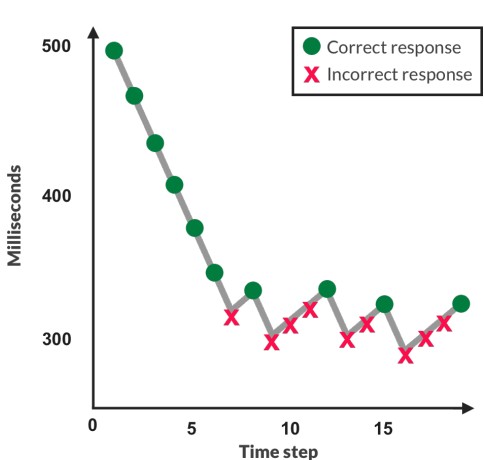

Every evaluator completes multiple staircases, called *blocks*, on different sets of images. As a result, we observe multiple measures for the model. We employ three blocks, to balance quality estimates against evaluators' fatigue (Krueger, 1989; Rzeszotarski et al., 2013). We average the modal exposure times across blocks to calculate a final value for each evaluator. Higher scores indicate a better model, whose outputs take longer time exposures to discern from real.

Figure 3: The adaptive staircase method shows images to evaluators at different time exposures, decreasing when correct and increasing when incorrect. The modal exposure measures their perceptual threshold. We repeat this method across multiple blocks, producing multiple staircases per evaluator per model.

## 2.2 Cost-effective approximation: HYPE$_\infty$

Building on the previous method, we introduce HYPE$_\infty$: a simpler, faster, and cheaper method after ablating HYPE$_{time}$ to optimize for speed, cost, and ease of interpretation. HYPE$_\infty$ shifts from a measure of perceptual time to a measure of human deception rate, given infinite evaluation time. The HYPE$_\infty$ score gauges total error on the task, enabling the measure to capture errors on both fake and real images, and effects of hyperrealistic generation when fake images look even more realistic than real images. HYPE$_\infty$ requires fewer images than HYPE$_{time}$ to find a stable value, at a 6x reduction in time and cost (10 minutes per evaluator instead of 60 minutes, at the same rate of \$12 per hour). Higher scores are better, like HYPE$_{time}$: a HYPE$_\infty$ value of 10% indicates that only 10% of images deceive people, whereas 50% indicates that people are mistaking real and fake images at chance, rendering fake images indistinguishable from real. Scores above 50% suggest hyperrealistic images, as evaluators mistake images at a rate greater than chance, on average mistaking more fake images to be real than real ones and vice versa.

HYPE$_\infty$ shows each evaluator a total of 100 images: 50 real and 50 fake. We calculate the proportion of images that were judged incorrectly, and aggregate the judgments over the $n$ evaluators on $k$ images to produce the final score for a given model.

## 2.3 Consistent and Reliable Design

To ensure that our reported scores are consistent and reliable, we need to sample sufficient model outputs, select suitable real images for comparison, and hire, qualify, and appropriately pay enough evaluators. To ensure a wide coverage of images, we randomly select the fake and real images provided to workers from a pool of 5000 images (see Sampling sufficient model outputs, below).

Comparing results between single evaluators can be problematic. To ensure HYPE is reliable, we must use a sufficiently large number of evaluators, $n$, which can be treated as a hyperparameter. To determine a suitable number, we use our experimental results (further discussed in the Results section) to compute bootstrapped $95\%$ confidence intervals (CI) across various values of $n$ evaluators.

**Quality of evaluators.** To obtain a high-quality pool of evaluators, each is required to pass a qualification task. Such a pre-task filtering approach, sometimes referred to as a person-oriented strategy, is known to outperform process-oriented strategies that perform post-task data filtering or processing (Mitra et al., 2015). Our qualification task displays 100 images (50 real and 50 fake) with no time limits. Evaluators pass if they correctly classify $65\%$ of both real and fake images. This threshold should be treated as a hyperparameter and may change depending upon the GANs used in the tutorial and the desired discernment ability of the chosen evaluators. We choose $65\%$ based on the cumulative binomial probability of 65 binary choice answers out of 100 total answers: there is only a one in one-thousand chance that an evaluator will qualify by random guessing. Unlike in the staircase task itself, fake qualification images are drawn equally from multiple different GANs. This is to ensure an equitable qualification across all GANs, as to avoid a qualification that is biased towards evaluators who are particularly good at detecting one type of GAN. The qualification is designed to be taken occasionally, such that a pool of evaluators can assess new models on demand.

**Payment.** Evaluators are paid a base rate of $1 for working on the qualification task. To incentivize evaluators to remained engaged throughout the task, all further pay after the qualification comes from a bonus of $0.02 per correctly labeled image. This pay rate typically results in a wage of approximately $12 per hour, which is above a minimum wage in our local state.

**Sampling sufficient model outputs.** The selection of $K$ images to evaluate from a particular model is a critical component of a fair and useful evaluation. We must sample a large enough number of images that fully capture a model's generative diversity, yet balance that against tractable costs in the evaluation. We follow existing work on evaluating generative output by sampling $K = 5000$ generated images from each model (Salimans et al., 2016; Miyato et al., 2018; Warde-Farley & Bengio, 2016) and $K = 5000$ real images from the training set. From these samples, we randomly select images to give to each evaluator.

# 3 EXPERIMENTS

**Datasets.** We evaluate on two datasets of human faces:

1. CelebA-64 (Liu et al., 2015) is popular dataset for unconditional image generation, used since 2015. CelebA-64 includes 202,599 images of human faces, which we align and crop to be $64 \times 64$ pixel images using a standard mechanism. We train all models without using attributes.

2. FFHQ-1024 (Karras et al., 2018) is a newer dataset released in 2018 with StyleGAN and includes 70,000 images of size $1024 \times 1024$ pixels.

**Architectures.** We evaluate on four state-of-the-art models trained on CelebA-64: StyleGAN (Karras et al., 2018), ProGAN (Karras et al., 2017), BEGAN (Berthelot et al., 2017), and WGAN-GP (Gulrajani et al., 2017). We also evaluate on two types of sampling from StyleGAN trained on FFHQ-1024: with and without the truncation trick, which we denote $\text{StyleGAN}_{\text{trunc}}$ and $\text{StyleGAN}_{\text{no-trunc}}$ respectively. For parity on our best models across datasets, StyleGAN trained on CelebA-64 is sampled with the truncation trick.

We train StyleGAN, ProGAN, BEGAN, and WGAN-GP on CelebA-64 using 8 Tesla V100 GPUs for approximately 5 days. We use the official released pretrained StyleGAN model on FFHQ-1024 (Karras et al., 2018).

We sample noise vectors from the $d$-dimensional spherical Gaussian noise prior $z \in \mathbb{R}^d \sim \mathcal{N}(0, I)$ during training and test times. We specifically opted to use the same standard noise prior for comparison, yet are aware of other priors that optimize for FID and IS scores (Brock et al., 2018). We select training hyperparameters published in the corresponding papers for each model.

We evaluate all models for each task with the two HYPE methods: (1) $\text{HYPE}_{\text{time}}$ and (2) $\text{HYPE}_\infty$.

**Evaluator recruitment.** We recruit 360 total human evaluators across our 12 evaluations, each of which included 30 evaluators, from Amazon Mechanical Turk. Each completed a single evaluation in {CelebA-64, FFHQ-1024} $\times$ {HYPE$_\text{time}$, HYPE$_\infty$}. To maintain a between subjects study in this evaluation, we did not allow duplicate evaluators across tasks or methods.

In total, we recorded (4 CelebA-64 + 2 FFHQ-1024) models $\times$ 30 evaluators $\times$ 550 responses = $99,000$ total responses for our HYPE$_\text{time}$ evaluation and (4 CelebA-64 + 2 FFHQ-1024) models $\times$ 30 evaluators $\times$ 100 responses = $18,000$ total responses for our HYPE$_\infty$ evaluation.

**Metrics.** For HYPE$_\text{time}$, we report the modal perceptual threshold in milliseconds. For HYPE$_\infty$, we report the error rate as a percentage of images, as well as the breakdown of this rate on real and fake images individually. To show that our results for each model are separable, we report a one-way ANOVA with Tukey pairwise post-hoc tests to compare all models within each {CelebA-64, FFHQ-1024} $\times$ {HYPE$_\text{time}$, HYPE$_\infty$} combination.

As mentioned previously, reliability is a critical component of HYPE, as an evaluation is not useful if a researcher can re-run it and get a different answer. To show the reliability of HYPE, we use bootstrap (Felsenstein, 1985), a form of simulation, to simulate what the results would be if we resample with replacement from this set of labels. Our goal is to see how much variation we may get in the outcome. We therefore report evaluator $95\%$ bootstrapped confidence intervals, along with standard deviation of the bootstrap sample distribution.

Confidence intervals (CIs) are defined as the region that captures where the modal exposure might be estimated to be if the same sampling procedure were repeated many times. For this and all following results, bootstrapped confidence intervals were calculated by randomly sampling 30 evaluators with replacement from the original set of evaluators across $10,000$ iterations. Note that bootstrapped CIs do not represent that there necessarily exists substantial uncertainty—our reported modal exposure (for HYPE$_\text{time}$) or detection rate (for HYPE$_\infty$) is still the best point estimate of the value. We discuss bootstrapped CIs for other numbers of evaluators later on in the Cost Tradeoffs section.

## 4 RESULTS

First, we report results using the above datasets, models and metrics using HYPE$_\text{time}$. Next, we demonstrate the HYPE$_\infty$'s results approximates the ones from HYPE$_\text{time}$ at a fraction of the cost and time. Next, we trade off the accuracy of our scores with time. We end with comparisons to FID.

### 4.1 HYPE$_\text{TIME}$

**CelebA-64.** We find that StyleGAN$_\text{trunc}$ resulted in the highest HYPE$_\text{time}$ score (modal exposure time), at a mean of 439.3ms, indicating that evaluators required nearly a half-second of exposure to accurately classify StyleGAN$_\text{trunc}$ images (Table 1). StyleGAN$_\text{trunc}$ is followed by ProGAN at 363.7ms, a $17\%$ drop in time. BEGAN and WGAN-GP are both easily identifiable as fake, so they are tied in third place around the minimum possible exposure time available of 100ms. Both BEGAN and WGAN-GP exhibit a bottoming out effect — reaching our minimum time exposure of 100ms quickly and consistently[1]. This means that humans can detect fake generated images at 100ms and possibly lower. Thus, their scores are identical and indistinguishable.

To demonstrate separability between StyleGAN$_\text{trunc}$, ProGAN, BEGAN, and WGAN-GP together, we report results from a one-way analysis of variance (ANOVA) test between all four models, where each model's input is the list of modes from each model's 30 evaluators. The ANOVA results confirm that there is a statistically significant omnibus difference ($F(3, 29) = 83.5, p < 0.0001$). Pairwise post-hoc analysis using Tukey tests confirms that all pairs of models are separable (all $p < 0.05$), with the exception of BEGAN and WGAN-GP ($n.s.$).

**FFHQ-1024**. We find that StyleGAN$_\text{trunc}$ resulted in a higher exposure time than StyleGAN$_\text{no-trunc}$, at 363.2ms and 240.7ms, respectively (Table 2). While the $95\%$ confidence intervals that represent a very conservative overlap of 2.7ms, an unpaired t-test confirms that the difference between the two models is significant ($t(58) = 2.3, p = 0.02$).

---

[1]We do not pursue time exposures under 100ms due to constraints on JavaScript browser rendering times.

| Rank | GAN | HYPE$_{time}$ (ms) | Std. | 95% CI |
|------|-----|------------------|------|--------|
| 1 | StyleGAN$_{trunc}$ | 439.4 | 35.7 | 306.7 – 400.8 |
| 2 | ProGAN | 363.7 | 29.6 | 227.0 – 304.7 |
| 3 | BEGAN | 111.1 | 8.7 | 100.0 – 130.3 |
| 3 | WGAN-GP | 100.0 | 0.0 | 100.0 – 100.0 |

Table 1: HYPE$_{time}$ on four GANs, trained on CelebA-64. Evaluators required the longest exposure times to distinguish StyleGAN$_{trunc}$ images, followed by ProGAN, and a tie between BEGAN and WGAN-GP. Note that the CI width for WGAN-GP is zero because the modal time exposure for all evaluators was 100ms, the lowest time displayed.

| Rank | GAN | HYPE$_{time}$ (ms) | Std. | 95% CI |
|------|-----|------------------|------|--------|
| 1 | StyleGAN$_{trunc}$ | 363.2 | 32.1 | 300.0 – 424.3 |
| 2 | StyleGAN$_{no-trunc}$ | 240.7 | 29.9 | 184.7 – 302.7 |

Table 2: HYPE$_{time}$ on StyleGAN$_{trunc}$ and StyleGAN$_{no-trunc}$ trained on FFHQ-1024. Evaluators required the longest time exposure to distinguish StyleGAN$_{trunc}$ images, thus making StyleGAN$_{trunc}$ generation significantly more realistic than that of StyleGAN$_{no-trunc}$.

## 4.2 HYPE$_\infty$

**CelebA-64**. Table 3 reports results for HYPE$_\infty$ on CelebA-64. We find that StyleGAN$_{trunc}$ resulted in the highest HYPE$_\infty$ score, fooling evaluators 50.7% of the time. StyleGAN$_{trunc}$ is followed by ProGAN at 40.3%, BEGAN at 10.0%, and WGAN-GP at 3.8%. No confidence intervals are overlapping and an ANOVA test is significant ($F(3, 29) = 404.4, p < 0.001$). Pairwise post-hoc Tukey tests show that all pairs of models are separable (all $p < 0.05$). Notably, HYPE$_\infty$ results in separable results for BEGAN and WGAN-GP, unlike in HYPE$_{time}$ where they were not separable due to a bottoming-out effect.

| Rank | GAN | HYPE$_\infty$ (%) | Fakes Error | Reals Error | Std. | 95% CI |
|------|-----|------------------|-------------|-------------|------|--------|
| 1 | StyleGAN$_{trunc}$ | 50.7% | 62.2% | 39.3% | 1.3 | 48.2 – 53.1 |
| 2 | ProGAN | 40.3% | 46.2% | 34.4% | 0.9 | 38.5 – 42.0 |
| 3 | BEGAN | 10.0% | 6.2% | 13.8% | 1.6 | 7.2 – 13.3 |
| 4 | WGAN-GP | 3.8% | 1.7% | 5.9% | 0.6 | 3.2 – 5.7 |

Table 3: HYPE$_\infty$ on four GANs trained on CelebA-64. Evaluators were deceived most often by StyleGAN$_{trunc}$ images, followed by ProGAN, BEGAN, and WGAN-GP. We also display the breakdown of the deception rate on real and fake images individually; counterintuitively, real errors increase with the errors on fake images, because evaluators become more confused and distinguishing factors between the two distributions become harder to discern.

**FFHQ-1024**. We observe a consistently separable difference between StyleGAN$_{trunc}$ and StyleGAN$_{no-trunc}$ and clear delineations between models (Table 4). HYPE$_\infty$ ranks StyleGAN$_{trunc}$ (27.6%) above StyleGAN$_{no-trunc}$ (19.0%) with no overlapping CIs. Separability is confirmed by an unpaired t-test ($t(58) = 8.3, p < 0.001$).

## 4.3 Cost tradeoffs with accuracy and time

One of HYPE's goals is to be cost and time efficient. When running HYPE, there is an inherent tradeoff between accuracy and time, as well as between accuracy and cost. This is driven by the law of large numbers: recruiting additional evaluators in a crowdsourcing task often produces more consistent results, but at a higher cost (as each evaluator is paid for their work) and a longer amount of time until completion (as more evaluators must be recruited and they must complete their work).

To manage this tradeoff, we run an experiment with HYPE$_\infty$ on StyleGAN$_{trunc}$. We completed an additional evaluation with 60 evaluators, and compute 95% bootstrapped confidence intervals,

| Rank | GAN | HYPE$_\infty$ (%) | Fakes Error | Reals Error | Std. | 95% CI |
|------|-----|-------------------|-------------|-------------|------|--------|
| 1 | StyleGAN$_{trunc}$ | 27.6% | 28.4% | 26.8% | 2.4 | 22.9 – 32.4 |
| 2 | StyleGAN$_{no-trunc}$ | 19.0% | 18.5% | 19.5% | 1.8 | 15.5 – 22.4 |

Table 4: HYPE$_\infty$ on StyleGAN$_{trunc}$ and StyleGAN$_{no-trunc}$ trained on FFHQ-1024. Evaluators were deceived most often by StyleGAN$_{trunc}$. Similar to CelebA-64, fake errors and real errors track each other as the line between real and fake distributions blurs.

choosing from 10 to 120 evaluators (Figure 4). We see that the CI begins to converge around 30 evaluators, our recommended number of evaluators to recruit and the default that we build into our system.

Payment to evaluators was calculated as described in the Approach section. At 30 evaluators, the cost of running HYPE$_{time}$ on one model was approximately $360, while the cost of running HYPE$_\infty$ on the same model was approximately $60. Payment per evaluator for both tasks was approximately $12/hr, and evaluators spent an average of one hour each on a HYPE$_{time}$ task and 10 minutes each on a HYPE$_\infty$ task. Thus, HYPE$_\infty$ achieves its goals of being significantly cheaper to run than HYPE$_{time}$ while maintaining consistency.

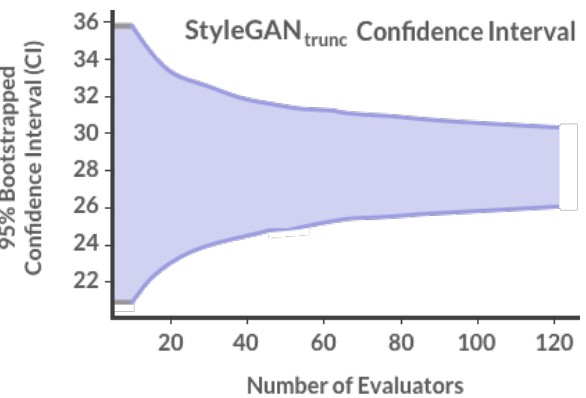

### 4.4 COMPARISON TO FID

As FID is one of the most frequently used evaluation methods for unconditional image generation, it is imperative to compare HYPE against FID on the same models (Table 5). We show through Spearman

Figure 4: We completed an additional evaluation of HYPE$_\infty$ on FFHQ-1024 with 60 evaluators, and compute 95% bootstrapped confidence intervals, choosing from 10 to 120 evaluators. We see that the CI begins to converge around 30 evaluators.

rank-order correlation coefficients that FID is correlated with neither human judgment measure, not HYPE$_{time}$ ($\rho = -0.0286$) nor with HYPE$_\infty$ ($\rho = -0.0857$), where a Spearman correlation of -1.0 is ideal because lower FID and higher HYPE scores indicate stronger models. Meanwhile, HYPE$_{time}$ and HYPE$_\infty$ exhibit strong correlation ($\rho = 0.9429$), where 1.0 is ideal because they are directly related. We calculate FID across the standard protocol of evaluating 50K generated and 50K real images for both CelebA-64 and FFHQ-1024, reproducing scores for StyleGAN$_{no-trunc}$.

| Metric | WGAN-GP | BEGAN | ProGAN | StyleGAN$_{trunc}$ | StyleGAN$_{trunc}$ | StyleGAN$_{no-trunc}$ |
|--------|---------|-------|--------|--------------------|--------------------|------------------------|
| HYPE$_{time}$ | 100.0 | 111.1 | 363.7 | 439.4* | 363.2* | 240.7 |
| HYPE$_\infty$ | 3.8 | 10.0 | 40.3 | 50.7* | 27.6* | 19.0 |
| FID | 43.6 | 67.7 | 2.5* | 131.7 | 13.8 | 4.4* |

Table 5: HYPE scores compared to FID. We put an asterisk on the most realistic GAN for each score (lower the better for FID, higher the better for HYPE). FID scores do not correlate fully with the human evaluation scores of HYPE$_\infty$ on both CelebA-64 and FFHQ-1024 tasks. FID scores were calculated using 50K real (CelebA-64 or FFHQ-1024) and 50K generated images for each model.

## 5 RELATED WORK

**Cognitive psychology.** We leverage decades of cognitive psychology to motivate how we use stimulus timing to gauge the perceptual realism of generated images. It takes an average of 150ms of focused visual attention for people to process and interpret an image, but only 120ms to respond to

faces because our inferotemporal cortex has dedicated neural resources for face detection (Rayner et al., 2009; Chellappa et al., 2010). Perceptual masks are placed between a person's response to a stimulus and their perception of it to eliminate post-processing of the stimuli after the desired time exposure (Sperling, 1963). Prior work in determining human perceptual thresholds (Greene & Oliva, 2009) generates masks from their test images using the texture-synthesis algorithm (Portilla & Simoncelli, 2000). We leverage this literature to establish feasible lower bounds on the exposure time of images, the time between images, and the use of noise masks.

**Success of automatic metrics.** Common generative modeling tasks include realistic image generation (Goodfellow et al., 2014), machine translation (Bahdanau et al., 2014), image captioning (Vinyals et al., 2015), and abstract summarization (Mani, 1999), among others. These tasks often resort to automatic metrics like the Inception Score (IS) (Salimans et al., 2016) and Fréchet Inception Distance (FID) (Heusel et al., 2017) to evaluate images and BLEU (Papineni et al., 2002), CIDEr (Vedantam et al., 2015) and METEOR (Banerjee & Lavie, 2005) scores to evaluate text. While we focus on how realistic generated content appears, other automatic metrics also measure diversity of output, overfitting, entanglement, training stability, and computational and sample efficiency of the model (Borji, 2018; Lucic et al., 2018; Barratt & Sharma, 2018). Our metric may also capture one aspect of output diversity, insofar as human evaluators can detect similarities or patterns across images. Our evaluation is not meant to replace existing methods but to complement them.

**Limitations of automatic metrics.** Prior work has asserted that there exists coarse correlation of human judgment to FID (Heusel et al., 2017) and IS (Salimans et al., 2016), leading to their widespread adoption. Both metrics depend on the Inception v3 Network (Szegedy et al., 2016), a pretrained ImageNet model, to calculate statistics on the generated output (for IS) and on the real and generated distributions (for FID). The validity of these metrics when applied to other datasets has been repeatedly called into question (Barratt & Sharma, 2018; Rosca et al., 2017; Borji, 2018; Ravuri et al., 2018). Perturbations imperceptible to humans alter their values, similar to the behavior of adversarial examples (Kurakin et al., 2016). Finally, similar to our metric, FID depends on a set of real examples and a set of generated examples to compute high-level differences between the distributions, and there is inherent variance to the metric depending on the number of images and which images were chosen—in fact, there exists a correlation between accuracy and budget (cost of computation) in improving FID scores, because spending a longer time and thus higher cost on compute will yield better FID scores (Lucic et al., 2018). Nevertheless, this cost is still lower than paid human annotators per image.

**Human evaluations.** Many human-based evaluations have been attempted to varying degrees of success in prior work, either to evaluate models directly (Denton et al., 2015; Olsson et al., 2018) or to motivate using automated metrics (Salimans et al., 2016; Heusel et al., 2017). Prior work also used people to evaluate GAN outputs on CIFAR-10 and MNIST and even provided immediate feedback after every judgment (Salimans et al., 2016). They found that generated MNIST samples have saturated human performance—that is, people cannot distinguish generated numbers from real MNIST numbers, while still finding 21.3% error rate on CIFAR-10 with the same model (Salimans et al., 2016). This suggests that different datasets will have different levels of complexity for crossing realistic or hyper-realistic thresholds. The closest recent work to ours compares models using a tournament of discriminators (Olsson et al., 2018). While reported to have anecdotal correlation with human judgment, this comparison was not yet rigorously evaluated and human discriminators were not presented experimentally. The framework we present would enable such a tournament evaluation to be performed reliably and easily.

# 6 DISCUSSION AND FUTURE WORK

We develop HYPE (1) **grounded** in psychophysics to measure human perceptual fidelity. Through empirical analysis, we find that (2) HYPE is **reliable** in its ability to replicate human perceptual performance measures across models and tasks, demonstrating non-overlapping 95% bootstrapped confidence intervals across all models with 30 evaluators. (3) HYPE also produces **separable** results and ranks models consistently using two methods: $\text{HYPE}_\text{time}$ and $\text{HYPE}_\infty$. $\text{HYPE}_\infty$ builds on $\text{HYPE}_\text{time}$ to (4) optimize for cost and time, enabling an efficient evaluation of a single model to occur within 10 minutes at $60.

**Envisioned Use.** We created HYPE to be a simple and rapid method of human evaluation of generative models. We envision that researchers will use HYPE via https://hype.stanford.edu to upload their model, receive a score, and compare progress. We also envision that HYPE, and specifically the variant $HYPE_\infty$ or $HYPE_{time}$ that fits best, to be used in generative model competitions, e.g. a $HYPE_\infty$ leaderboard on CelebA-64. Competition organizers can have the license to choose whether the model alone should be compared, or that automatic sampling methods such as the truncation trick should be included as part of the system.

During periods of high usage, such as competitions, a retainer model (Bernstein et al., 2011) enables nearly instantaneous availability of evaluators, meaning evaluation using $HYPE_\infty$ could take as few as 10 minutes. Without the retainer model, however, we estimate that time rises to 30 minutes.

**Limitations.** Extensions of HYPE may require different task designs. For example, conditional image generation will need to isolate different categories into individual blocks (e.g. cats in one block, dogs in another, etc.) (Krishna et al., 2016). The design would likely affect humans' absolute thresholds, as cognitive load may be of consideration; the number of humans required per task may require significant increase if evaluating fairly across all possible categories. Practically, the most valuable direction for the community to pursue with HYPE is likely one that includes the most difficult categories, especially when progress on those is hard to measure using automatic metrics. In the case of text generation (translation, caption generation), $HYPE_{time}$ may require much longer and much higher range adjustments to the perceptual time thresholds for text comprehensibility than those used in visual perception (Krishna et al., 2016).

**Future Work.** We plan to extend HYPE to different imaging datasets and imaging tasks such as conditional image generation, as well as to text and video, such as translation (Papineni et al., 2002) and video captioning (Krishna et al., 2017). Future work would also explore budget-optimal estimation of HYPE scores and adaptive estimation of evaluator quality (Karger et al., 2014). Additional improvements involve identifying images that require more evaluators (Weld et al., 2015). We also aim to build in faster time exposures under 100ms — ideally down to 13ms, the minimum time exposure of human perception (Potter et al., 2014) — for tasks that require that level of granularity. Doing so requires careful engineering solution, since 100ms appears to be the minimum time that is trustable before we are throttled by JavaScript paint and rendering times on modern browsers.

We will investigate the ecological validity of our methods – that is, whether HYPE's evaluation is representative of how a person would perceive a GAN in everyday life. For instance, HYPE shows evaluators whether they classified an image correctly immediately after they answer. While this is standard practice in the psychophysics literature for staircase tasks, it likely does not reflect how one might encounter generated content in everyday life. Notably, in pilot studies, we found that without such feedback, evaluators were far less consistent and our metric would not be stable.

Finally, we plan to investigate whether the reliability of HYPE may be impacted by the month or year at which it is run, as the population of available crowdsourced workers may differ across these factors. Anecdotally, we have found HYPE to be reliable regardless of the time of day.

## 7  CONCLUSION

HYPE provides researchers with two human evaluation methods for GANs that (1) are **grounded** in psychopisics to measure human perceptual fidelity directly, (2) provide task designs that result in consistent and **reliable** results, (3) distinguishes between different model performances through **separable** results, (4) is cost and time **efficient**. We report two metrics: $HYPE_{time}$ and $HYPE_\infty$. $HYPE_{time}$ uses time perceptual thresholds where longer time constraints are more difficult to achieve because they give humans more time to interpret the generated content and observe artifacts. $HYPE_\infty$ reports the error rate under unlimited time, where higher rates indicate a more realistic set of outputs. We demonstrate the efficacy of our approach on unconditional image generation across four GANs {StyleGAN, ProGAN, BEGAN, WGAN-GP} and two datasets of human faces {CelebA-64, FFHQ-1024}, with two types of output sampling on StyleGAN {with the truncation trick, without the truncation trick}. To encourage progress of generative models towards human-level visual fidelity, we deploy our evaluation system at https://hype.stanford.edu, so anyone can upload and evaluate their models based on HYPE at the click of a button.

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

APPENDIX

A. CONFIDENCE INTERVALS

Table 6: Full bootstrapped 95% confidence intervals from $HYPE_\infty$

| Model | Std | Deception Rate | 95% CI | Number of Evaluators |
|---|---|---|---|---|
| WGAN-GP | 0.84 | 3.79 | 2.29–5.52 | 10 |
| WGAN-GP | 0.69 | 3.78 | 2.5–5.21 | 15 |
| WGAN-GP | 0.61 | 3.78 | 2.66–5.05 | 20 |
| WGAN-GP | 0.54 | 3.79 | 2.79–4.92 | 25 |
| WGAN-GP | 0.49 | 3.78 | 2.85–4.76 | 30 |
| WGAN-GP | 0.45 | 3.78 | 2.95–4.7 | 35 |
| WGAN-GP | 0.43 | 3.79 | 2.97–4.64 | 40 |
| BEGAN | 2.65 | 10.04 | 5.73–15.83 | 10 |
| BEGAN | 2.20 | 9.99 | 6.18–14.72 | 15 |
| BEGAN | 1.86 | 9.97 | 6.72–13.91 | 20 |
| BEGAN | 1.67 | 10.01 | 7.04–13.62 | 25 |
| BEGAN | 1.55 | 9.99 | 7.26–13.3 | 30 |
| BEGAN | 1.42 | 10.00 | 7.44–12.95 | 35 |
| BEGAN | 1.33 | 10.00 | 7.6–12.76 | 40 |
| ProGAN | 1.53 | 40.30 | 37.19–43.23 | 10 |
| ProGAN | 1.24 | 40.28 | 37.78–42.57 | 15 |
| ProGAN | 1.08 | 40.29 | 38.12–42.4 | 20 |
| ProGAN | 0.97 | 40.27 | 38.38–42.12 | 25 |
| ProGAN | 0.87 | 40.29 | 38.54–41.94 | 30 |
| ProGAN | 0.83 | 40.27 | 38.6–41.87 | 35 |
| ProGAN | 0.76 | 40.29 | 38.75–41.74 | 40 |
| StyleGAN | 2.19 | 50.66 | 46.35–54.9 | 10 |
| StyleGAN | 1.79 | 50.68 | 47.08–54.17 | 15 |
| StyleGAN | 1.57 | 50.67 | 47.6–53.7 | 20 |
| StyleGAN | 1.38 | 50.65 | 47.92–53.29 | 25 |
| StyleGAN | 1.27 | 50.66 | 48.16–53.16 | 30 |
| StyleGAN | 1.16 | 50.68 | 48.39–52.95 | 35 |
| StyleGAN | 1.09 | 50.64 | 48.49–52.71 | 40 |
| $StyleGAN_{trunc}$ | 4.21 | 27.59 | 19.7–35.9 | 10 |
| $StyleGAN_{trunc}$ | 3.46 | 27.60 | 20.93–34.53 | 15 |
| $StyleGAN_{trunc}$ | 2.95 | 27.65 | 22.0–33.55 | 20 |
| $StyleGAN_{trunc}$ | 2.69 | 27.58 | 22.44–32.96 | 25 |
| $StyleGAN_{trunc}$ | 2.44 | 27.60 | 22.9–32.43 | 30 |
| $StyleGAN_{trunc}$ | 2.25 | 27.64 | 23.34–32.17 | 35 |
| $StyleGAN_{trunc}$ | 2.13 | 27.62 | 23.5–31.87 | 40 |
| $StyleGAN_{no\text{-}trunc}$ | 3.04 | 19.06 | 13.1–25.1 | 10 |
| $StyleGAN_{no\text{-}trunc}$ | 2.50 | 19.02 | 14.13–23.93 | 15 |
| $StyleGAN_{no\text{-}trunc}$ | 2.17 | 18.99 | 14.75–23.25 | 20 |
| $StyleGAN_{no\text{-}trunc}$ | 1.96 | 18.96 | 15.08–22.84 | 25 |
| $StyleGAN_{no\text{-}trunc}$ | 1.77 | 18.95 | 15.5–22.43 | 30 |
| $StyleGAN_{no\text{-}trunc}$ | 1.62 | 18.98 | 15.8–22.23 | 35 |
| $StyleGAN_{no\text{-}trunc}$ | 1.52 | 18.98 | 16.0–22.05 | 40 |

Table 7: Full bootstrapped 95% confidence intervals from $\text{HYPE}_{\text{time}}$

| Model | Std | Mean | 95% CI | Number of Evaluators |
|---|---|---|---|---|
| WGAN-GP | 0.00 | 100.00 | 100.0–100.0 | 10 |
| WGAN-GP | 0.00 | 100.00 | 100.0–100.0 | 15 |
| WGAN-GP | 0.00 | 100.00 | 100.0–100.0 | 20 |
| WGAN-GP | 0.00 | 100.00 | 100.0–100.0 | 25 |
| WGAN-GP | 0.00 | 100.00 | 100.0–100.0 | 30 |
| BEGAN | 12.50 | 107.28 | 100.0–100.0 | 10 |
| BEGAN | 10.08 | 107.33 | 100.0–100.0 | 15 |
| BEGAN | 8.85 | 107.41 | 100.0–100.0 | 20 |
| BEGAN | 7.86 | 107.34 | 100.0–108.8 | 25 |
| BEGAN | 7.06 | 107.24 | 100.0–107.33 | 30 |
| $\text{StyleGAN}_{\text{trunc}}$ | 52.29 | 364.88 | 243.0–365.0 | 10 |
| $\text{StyleGAN}_{\text{trunc}}$ | 42.95 | 365.84 | 266.0–366.0 | 15 |
| $\text{StyleGAN}_{\text{trunc}}$ | 37.54 | 365.35 | 279.0–365.0 | 20 |
| $\text{StyleGAN}_{\text{trunc}}$ | 33.21 | 365.33 | 288.0–365.2 | 25 |
| $\text{StyleGAN}_{\text{trunc}}$ | 30.34 | 365.59 | 295.67–365.0 | 30 |
| $\text{StyleGAN}_{\text{trunc}}$ | 28.12 | 365.20 | 301.43–364.57 | 35 |
| $\text{StyleGAN}_{\text{trunc}}$ | 26.22 | 365.60 | 305.5–365.25 | 40 |
| $\text{StyleGAN}_{\text{no-trunc}}$ | 49.28 | 242.99 | 143.0–239.0 | 10 |
| $\text{StyleGAN}_{\text{no-trunc}}$ | 40.33 | 243.07 | 157.33–240.67 | 15 |
| $\text{StyleGAN}_{\text{no-trunc}}$ | 34.75 | 243.71 | 171.5–241.5 | 20 |
| $\text{StyleGAN}_{\text{no-trunc}}$ | 31.31 | 243.16 | 176.4–241.6 | 25 |
| $\text{StyleGAN}_{\text{no-trunc}}$ | 28.32 | 243.33 | 182.0–241.67 | 30 |
| $\text{StyleGAN}_{\text{no-trunc}}$ | 26.19 | 243.19 | 187.14–242.0 | 35 |
| $\text{StyleGAN}_{\text{no-trunc}}$ | 24.37 | 243.03 | 191.25–241.75 | 40 |
| ProGAN | 51.03 | 335.55 | 221.0–335.0 | 10 |
| ProGAN | 41.97 | 336.14 | 241.33–335.33 | 15 |
| ProGAN | 36.31 | 335.38 | 252.0–335.0 | 20 |
| ProGAN | 32.25 | 336.05 | 262.4–335.2 | 25 |
| ProGAN | 29.72 | 335.17 | 268.0–334.67 | 30 |
| StyleGAN | 56.36 | 368.82 | 237.0–368.0 | 10 |
| StyleGAN | 46.07 | 369.46 | 262.0–368.67 | 15 |
| StyleGAN | 39.94 | 368.86 | 277.0–368.5 | 20 |
| StyleGAN | 35.66 | 368.66 | 285.2–368.4 | 25 |
| StyleGAN | 32.53 | 368.82 | 295.33–368.33 | 30 |

