# OpenReview forum: "HYPE:  Human-eYe Perceptual Evaluation of Generative Models"
_ICLR.cc/2019/Workshop/DeepGenStruct — DeepGenStruct 2019_

### Official Review · AnonReviewer2 · 2019-04-15
**Timely work, well presented**

**Rating:** 4
**Confidence:** 2

**Review:**

The paper proposes a framework for human evaluation of generative models of images. It is based on samples, so it is compatible with any flavour of generative model (likelihood-based, adversarial or otherwise). Two different evaluation strategies are proposed: one based on the time it takes for humans to distinguish generated images from real images, and another which simply measures the percentage of images that are wrongly classified.

The implementation of the human evaluation setup is described in appropriate detail, and attention to cost is also given. The results are comprehensive and statistical tests are used to show their significance. The approach is also compared to FID, a computational evaluation metric that is currently popular.

Overall, this is work is timely and it is well presented, so I am in favour of acceptance. Nevertheless I have a few more comments and suggested improvements below:

* It is demonstrated that the correlation of HYPE and FID is relatively poor, and it is implied that this demonstrates that FID is a poor metric. However, as the authors state earlier on in the paper, HYPE can only measure realism, not diversity. FID is explicitly constructed to also be affected by sample diversity, so in that light it is not surprising that the two do not correlate very well, and that higher truncation leads to improved HYPE but worse FID scores -- it is well known that truncation reduces diversity of the samples, in favour of improved fidelity. (I do not wish to imply that FID is actually a good metric -- I do believe it is a poor metric, but not for this particular reason.)

* While the authors state clearly that HYPE does not measure diversity, I think it would be worth discussing in more detail how one could use human evaluation to measure diversity, as it is arguably a more interesting challenge. As it stands, the HYPE metric could probably be fooled by a "model" which simply stores a few training examples and randomly selects them with equal probability. Also measuring the diversity of the samples in some way would prevent this kind of cheating.

* A common issue with human evaluation is ambiguity in the task specification: the raters are instructed to determine which images are real, but they may be prone to misinterpreting the task in a way that biases the results. While rater training and immediate feedback undoubtedly help to limit this effect, it is still worth considering this carefully, and I think a few diagrams or screenshots of the rater interface would be useful additions to the manuscript in this respect.

* In the introduction, it is implied that likelihood (measured in the input space) would be the ideal metric for generative models if it were always easy to compute (which it often isn't). Theis et al. (2015), cited there, also call this into question. I find the juxtaposition of this citation and the sentence before it a bit misleading.

---

### Official Review · AnonReviewer1 · 2019-04-17
**Review for "HYPE: Human-eYe Perceptual Evaluation of Generative Models"**

**Rating:** 4
**Confidence:** 2

**Review:**

The paper introduces two methods for evaluating generative models, "HYPE_time" and "HYPE_infinity". Both methods use human reaction times or error rates when asked to distinguish fake images from real data.

Several comments about the paper:

Pros:
- The paper is very well written and clear.
- The process is fully automated and reproducible, which is a drastic difference with other user studies this reviewer is aware of. The process is thoroughly documented.
- This paper is quite out there, the ideas are to me very novel and reasonably surprising (at the degree of thoroughness that this paper makes use of them). I don't think this is ready for a conference, but in terms of giving a space for more risky ideas to be discussed (as workshops should), I think this paper fits the bill.
- As mentioned in the paper, the method seems to be reliable, moderately fast (10 minutes), and measures perceptual fidelity well.
- I think the experiments are enough evidence to support the authors' claims.

Cons:
- It's undiscussed (and to me unlikely) wether the proposed scores are good for ranking models that are bad (but one a lot worse than the other, such as models in the middle of training), given that distinguishability for bad models might be essentially the same, or require a lot more time and humans to reach confidence intervals that are non-overlapping.
- There is an emphasis on 'reality', which ignores 'diversity', and I think the paper should stress more that this in effect is not trying to provide a way to evaluate generative models per se, but simply the 'reality' of the samples.
- In the same lines of the above, 'reality' is not the same as 'fidelity', a model that for examples produces reasonable faces but ignores modeling the background of an image distribution has obviously worse sample quality, but a human might think the samples to be more real. Essentially, it focuses to much on the human-centric preconceived notion of reality, rather than a comparison (human based or not) between the true and generated data distributions.
- All experiments are on faces, which makes the reader wander wether the accuracy or 'cost-effectiveness' of the method depends on how good humans are at judging the quality of faces.
- It's quite expensive to run this thing ($60 makes it unusable for e.g. grid searches).
- The method is obviously specific to data types like images where humans have a notion of what a 'real' sample is, which limits the method from ever working in things like representation spaces.

---

### Decision · Program_Chairs · 2019-04-19
**Acceptance Decision**

Accept